

# A novel immune-related prognostic signature based on Chemoradiotherapy sensitivity predicts long-term survival in patients with esophageal squamous cell carcinoma

Zewei Zhang[1,2,*], Shiliang Liu[1,2,*], Tiantian Gao[1,2], Yuxian Yang[1,2], Quanfu Li[3] and Lei Zhao[1,2]

[1] Sun Yat-sen University Cancer Center, Guangzhou, China
[2] State Key Laboratory of Oncology in South China, Guangzhou, China
[3] Ordos Central Hospital, Ordos, China
[*] These authors contributed equally to this work.

Corresponding authors
Quanfu Li, 1729259137@qq.com
Lei Zhao, zhaolei@sysucc.org.cn

## ABSTRACT

**Background.** There is a heterogenous clinical response following chemoradiotherapy (CRT) in esophageal squamous cell carcinoma (ESCC). Therefore, we aimed to study signaling pathway genes that affect CRT sensitivity and prognosis.

**Methods.** Gene expression analyses were performed in the GEO and TCGA datasets. A immunohistochemistry (IHC) analysis was performed in pretreatment biopsies.

**Results.** MMP13 was found to be highly expressed in the "Pathologic Complete Response (pCR)" and "Complete Remission (CR)" and "Alive" groups. Th17 cells and MMP9/13 showed a negative correlation in immune infiltration analysis. In GSEA analysis, IL-4 and IL-13 signaling pathways were highly enriched in patients exhibiting high MMP expression in pCR and CR groups. IHC results suggested higher MMP13 & IL-4 and lower IL-17A & RORC expression in the CR group compared to the <CR (CR not achieved) group. Survival analyses further indicated that the prognosis was worse in the high IL-17A group ($p = 0.046$, HR = 2.15). Next, a prognostic model was established. In the training cohort, AUCs for the 1/2/3/4/5-year OS were all greater than 0.70. In the two validation cohorts, 1-year AUCs were also >0.70, and the model could well distinguish high-risk and low-risk subgroups.

**Conclusion.** The above results may provide guidance for developing novel treatment and prognostic strategies in ESCC patients.

# INTRODUCTION

Esophageal cancer (ESCA) ranks ninth in incidence among common cancers and sixth in terms of the number of cancer-related deaths (*Sung et al., 2021*). In 2020, 604,100 new

cases of ESCA and 544,076 related deaths were reported globally; approximately half of ESCA-related deaths occurred in China (*Arnold et al., 2020*). Based on histological subtypes, ESCA is classified into squamous cell carcinoma (ESCC) and adenocarcinoma (EAC) (*Lagergren et al., 2017*). ESCC is the most common subtype worldwide, particularly in East Asia, whereas EAC is the major subtype in western countries (*Lagergren et al., 2017*; *Morgan et al., 2022*). Although the 5-year survival rate of patients with ESCA has increased from 20.9% to 30.3% over the past decade, the prognosis of this cancer remains poor due to its late diagnosis (*Zeng et al., 2018*). Definitive chemoradiotherapy (dCRT) or neoadjuvant chemoradiotherapy (nCRT) are standard treatments for locally advanced ESCA (*Hulshof et al., 2021*; *Eyck et al., 2021*); however, there is a high level of heterogeneity in the degrees of clinical remission following CRT, including complete remission (CR), partial response (PR), stable disease and progressive disease (*Eyck et al., 2021*). After completion of nCRT, patients may achieve pathological complete response (pCR), which refers to the inability to detect any residual cancer cells under a microscope in resected specimens after treatment (*Sjödahl et al., 2022*). pCR is associated with improved prognosis and long-term survival. It indicates a high degree of treatment response, implying a lower risk of cancer recurrence. CR, on the other hand, means the absence of clinically detectable signs and symptoms of cancer, but does not necessarily correlate with long-term survival outcomes. Previous studies have found that patients who are sensitive to radiotherapy and chemotherapy have a better prognosis and longer overall survival (OS) (*Zhao et al., 2020*). Therefore, it is important to investigate the underlying molecular mechanism affecting the sensitivity or resistance to chemoradiotherapy, which could improve prognostic accuracy and elucidation of potential therapeutic targets.

Tumor microenvironment (TME) includes a variety of immune cells, tumor-associated fibroblasts (CAFs), endothelial cells (ECs), and extracellular matrix (ECM). TME plays a key role in regulating the immune response to cancer (*De Visser & Joyce, 2023*). Extensive remodeling of ECM often occurs during cancer progression, and ECM is degraded by the matrix metalloproteinase (MMP) family (*Wang et al., 2022*). MMPs are produced by a variety of tissues and cells in the TME, including cancer cells, CAFs, ECs, vascular smooth muscle, macrophages, neutrophils, and lymphocytes. It has also been reported that elevated MMP-13 levels are associated with tumor progression and poorer survival in ESCC patients (*Jiao et al., 2014*). However, researchers have found that MMPs may also slow tumorigenesis and cancer progression (*Nakasone et al., 2012*; *Fukuda et al., 2011*; *Decock et al., 2011*). By regulating vascular leakage, MMPs improve cancer drug sensitivity (*Nakasone et al., 2012*). In addition, endothelial cells oversecrete MMP-13, which could reduce tumor cell extravasation to suppress tumor metastasis through the local generation of endostatin (*Fukuda et al., 2011*). MMP1/3/9/13, the downstream molecules in the interleukin (IL)-17 signaling pathway, are activated by IL-17A/F (*Cortez et al., 2007*). Interestingly, in the field of ESCA, the role of IL-17 in tumor immunity is also controversial. On the one hand, in the ESCC TME, IL-17 can recruit beneficial neutrophils, CD8+ T lymphocytes and B lymphocytes to inhibit tumor growth (*Chen et al., 2017*; *Lu et al., 2016*). On the other hand, IL-17 has pro-tumorigenic effects which are achieved by directly promoting cellular proliferation and indirectly by creating an immunosuppressive TME (*Al-Samadi et al.,*

*2016*; *He et al., 2010*). In addition, IL-17A can significantly up-regulate the expression of MMP-2 and MMP-9 in EAC cells, stimulate the production of intracellular reactive oxygen species, thereby promoting the migration and invasion (*Liu et al., 2017*). Therefore, the expression levels of MMPs and IL-17A may be related to the efficacy and prognosis of ESCC. As potential therapeutic targets and immune-related prognostic factors, the mechanism of mutual regulation between them still needs further study.

*Wen et al. (2014)* built a three-gene CRT-response prediction model, including MMP1, LIMCH1 and C1orf226. Moreover, MMP1/9/12 were highly expressed in patients who achieved pCRs with a better response to nCRT than those who did not (*Wen et al., 2014*; *Zhang et al., 2020*). However, the researchers did not identify the signaling pathways involved in the signature genes, nor did they study immune cells that might affect CRT sensitivity. Now we have conducted an in-depth study on these two aspects. Our objective was to determine whether immune-related genes differed in ESCC patients with different treatment outcomes (pCRs/CRs and <pCRs/ <CRs (pCR/CR not achieved)) after receiving nCRT or dCRT. We constructed an immune-related signature based on signal pathway genes related to MMPs and IL-17A which engaged in CRT sensitivity, and it could predict long-term OS. Further, to explore possible gene correlation, we used immunohistochemistry (IHC) to verify the expression of these prognostic genes. Nevertheless, considering the increased recognition for the role of the immune system, we assumed that the model might offer an appealing platform for the identification of ESCC patients with both poor and good prognostic molecular subtypes.

## MATERIALS AND METHODS

### Flow chart and design of the study

Table S1 shows the clinical characteristics of the enrolled patients. It is important to note that the OS data of 28 patients in the GSE45670 microarray were retrieved from our center's follow-up system. Tables S2 & S3 show the clinical characteristics of patient cohorts in TCGA-ESCA "primary therapy outcome" dataset and the "OS event" dataset. Figure 1 presents a flow chart of the specific study.

### Gene Expression Omnibus (GEO) publicly available mRNA data

GEO is an open-access database that gathers microarray data from around the world (*Barrett et al., 2013*). In this study, in accordance with GEO accession number GSE45670, we downloaded 28 cases from Sun Yat-sen University Cancer Center (*Wen et al., 2014*). The dataset was comprised of clinical samples, including 11 pCRs and 17 <pCR, who received nCRT with locally advanced ESCC.

### The Cancer Genome Atlas (TCGA) data acquisition

TCGA is a project devoted to cancer genomics consisting of over 2.5 PB of genome data (*Tomczak, Czerwińska & Wiznerowicz, 2015*). Clinical information from ESCA projects coupled with gene expression data (including 11 normal and 162 tumor tissues, workflow type: HTSeq-FPKM) were obtained. The normal ESCA samples as well as patients who have insufficient clinical features or "0" gene expression values were excluded. The data is summarized in Table S4.

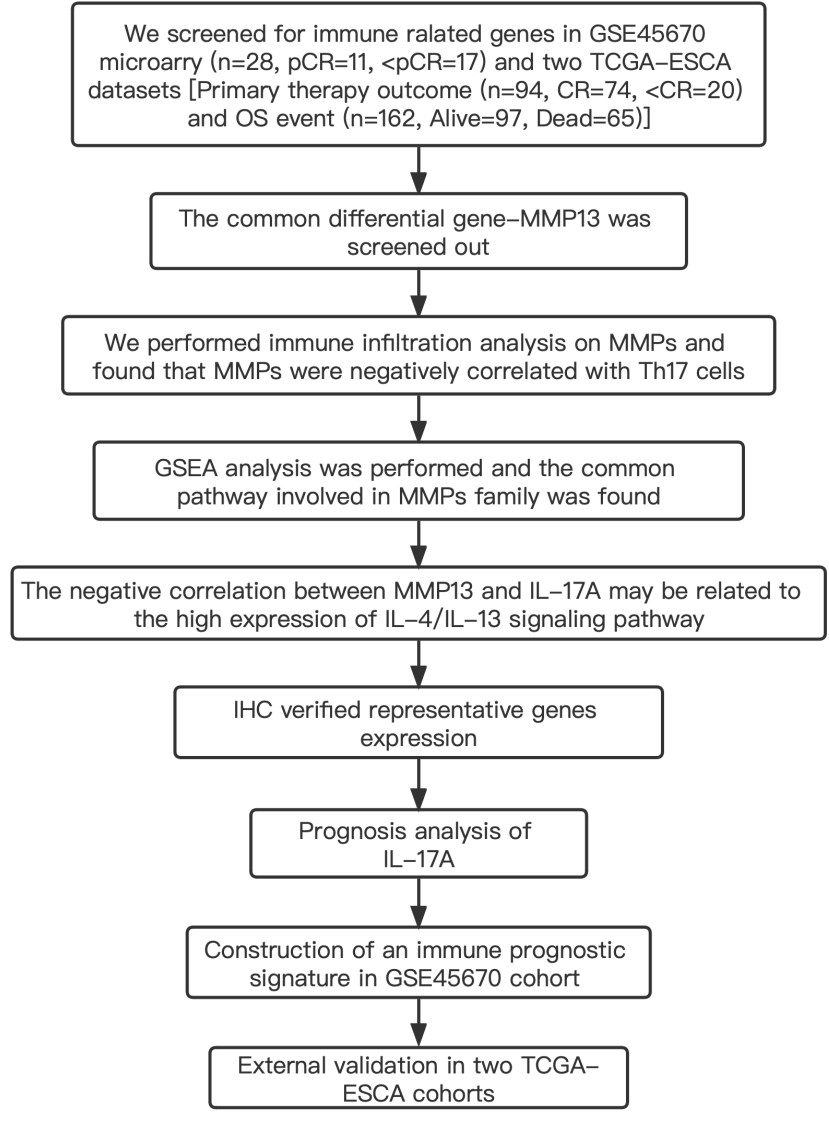

**Figure 1   Study flow.** Two institutions participated in the study, including Guangzhou (Sun Yat-sen University Cancer Center), and Ordos (Ordos Central Hospital). We followed up 28 patients in the GSE45670 cohort until December 2021. IHC, immunohistochemistry; CR, complete remission; <CR, CR not achieved; pCR, pathological complete response; <pCR, pCR not achieved.

## Analysis of immune infiltration using single-sample geneset enrichment analysis (ssGSEA)

In order to examine the immune infiltration landscape of esophageal cancer, using ssGSEA, immunoinfiltration was evaluated based on immune cell-specific markers genes expressing. We downloaded and sorted the RNAseq data of the STAR process of the TCGA-ESCA project from the TCGA database (https://portal.gdc.cancer.gov) and extracted the data in TPM format and clinical data. There were 174 cases with RNAseq; 11 cases with adjacent tumors; 185 cases with clinical data (with clinical information but no corresponding

RNAseq); 174 cases with RNAseq containing clinical information; one case with RNAseq data from the same patient. Log2 (value + 1) transformation is performed on the data after removing normal samples. We performed a correlation analysis between MMPs and immune infiltration matrix data, and the results were visualized in lollipop plots using the ggplot2 package [3.3.6]. The statistical method is Spearman. The immune infiltration algorithm is based on the ssGSEA algorithm provided in the R package-GSVA [1.46.0] (*Hänzelmann, Castelo & Guinney, 2013*). We used the markers of 24 kinds of immune cells provided by the Immunity article (*Bindea et al., 2013*) to calculate the immune infiltration of 174 cases of ESCAs. Next, we analyzed the difference in the level of immune infiltration of TH17 cells between the MMP13 high and low expression groups (grouped by the median). Because two sets of independent data meet the normality test and variance homogeneity test, we chose T test for statistics (stats package (4.2.1) and car package (3.1-0)), and used ggplot2 to visualize. The polygenic correlation heatmap is displayed by the pheatmap package. Estimation scores of immune infiltration level were downloaded in TIMER 2.0 (http://timer.cistrome.org/). TIMER2.0 uses the R immunedeconv package, which integrates four latest algorithms, including TIMER, MCP-counter, EPIC and quantIseq (*Li et al., 2020*). MCP-counter is a method to quantify tumor immune cells, fibroblasts and epithelial cells based on marker gene sets.

## Histological examination

ESCC paraffin sections were obtained from patients who received neoadjuvant or definitive chemoradiotherapy at the Department of Oncology, Ordos Central Hospital from January 2018 to December 2021. IHC was performed to analyze the differential expression of IL-4, MMP3, MMP13, RORA, RORC, and IL-17A among patients who achieved CR (three patients) or non-CR (three patients) following nCRT or dCRT with independent *t* test. Sections 3 mm in thickness were incubated with rabbit polyclonal antibodies against MMP3, MMP13, IL-4 (all from Cloud-Clone Corp., Houston, TX, USA), IL-17A (PA00406HuA10; Biolight, Zhuhai, China), RORA (ab70061; Abcam, Cambridge, UK), and RORC (ab219501; Abcam, Cambridge, UK) at 1/100 dilution overnight at 4 °C. We discarded the primary antibody and washed the sections with PBS for 10 min × 3 times. The goat anti-rabbit secondary antibody was added dropwise, and was incubated in a wet box at 37 °C for 20 min. Then it was discarded, and the sections were soaked in PBS for 5 min × 3. Then, the DAB working solution was added dropwise. After the color development was completed, the slices were soaked with ddH2O for 5 min × 3. Then, we counterstained with hematoxylin for 1–3 s, and washed with tap water for 10 min. For dehydration, we used 60% alcohol for 1s-3s, 90% alcohol for 1s-3s, 100% alcohol for 1 min × 2, and environmental transparent agent for 2 min × 3. Finally, the slides were mounted with ultra-clean mounting medium. Analysis of the IHC mean optical density was conducted using Image Pro Plus 6.0. For each slice, at least three 200 times visual fields were randomly selected and photographed. To achieve consistent background lighting, the entire field of vision was filled with the organization while photos were taken. All photopositive cells were judged using the same brown-yellow criterion, and each photo

was measured for integrated optical density (IOD) and pixel area (area). Finally, the mean optical density IOD/area was calculated.

## Lasso coefficient identification

The least absolute shrinkage and selection operator (Lasso) penalty was applied in the GSE45670 training cohort to build an optimal prognostic signature with the minimum number of signaling molecules. Prognostic Lasso is often used to construct prognostic models or screen variables. When there are fewer samples or more variables (variables less than half of the samples), Lasso can be used to directly build a prognosis model or screen variables (*Zhou et al., 2019*; *Sauerbrei, Boulesteix & Binder, 2011*; *Tibshirani, 1996*). To tune the optimal penalty parameter lamba, which yields the minimum partial likelihood deviance, the glmnet package (v4.1-2) and survival package (v3.2-10) were used to conduct a 10-fold cross validation. MMP1/2/3/9/10/13, CCL11, IL4/13, and IL17A were screened. Using cv.glmnet function, tenfold cross-validation was performed to find lambda minimum that gave the smallest cross-validated error. The original output of Cv.glmnet function is: Lambda.min = 0.1124, Index = 10, Mearsure = 7.456, SE = 0.4798, Nonzero = 5. Using lambda.min as a cutoff, five variables with non-zero coefficients were screened out, which could then be incorporated into the lasso prognosis model.

## Statistical analysis

Statistical analyses and figures were produced using the program R, version 3.6.3, as well as SPSS 26.0 (SPSS, Chicago, IL). Limma package (v3.42.2) in GSE45670 dataset and DESeq2 package (v1.26.0) in TCGA-ESCA "Primary therapy outcome" and "OS event" datasets for DEGs computations, and we used multiple comparison corrections with Benjamin and Hochberg FDR (BH) and got adjusted $p$ value (*Smyth, 2005*; *Love, Huber & Anders, 2014*). We chose adjusted $P < 0.05$ & log $|FC| > 1.5$ as DEGs' cutoff value in the two TCGA datasets. However, in GSE45670 dataset, since no differential genes were screened out with adj. $p$.val $< 0.05$ (Table S4), and the $p$-value $< 0.05$ is also allowed in the bioinformatics analysis (*Cheng et al., 2021*; *Shi et al., 2021*), we use $p$.val $< 0.05$ & log $|FC| > 1.5$ as the cutoff. Performance of the model was evaluated through timeROC package (v0.4). Survival package (v3.2-10) was used for survival analysis, K-M curve and Log rank test were used to assess survival difference between two patient subsets. For dividing the patients into low and high risk subsets, the "surv_cutpoint" function in the R package "survminer" (v0.4.9) was used to calculate the best cutoff risk score value. $P < 0.05$ was considered statistically significant, and all $P$ were bilateral.

## Ethics approval and consent to participate

The study was conducted according to the guidelines of the Declaration of Helsinki. The study protocol and the use of human samples had the ethics committee approval of Sun Yat-sen University Cancer Center and Ordos Central Hospital (ethics numbers: SL-B2022-054-01). Patient consent was waived due to the patient data was de-identified.

## RESULTS

### Identification of differentially expressed genes (DEGs) in ESCA patients after receiving CRT

The GSE45670 microarray from Sun Yat-sen University Cancer Center was selected for this research. This database is used to study the differences in gene expression from 28 pretreatment biopsies of ESCCs who have reached pCR and non-pCR following nCRT. Next, to screen out common DEGs in other databases, we performed a DEG analysis in two different datasets from the TCGA esophageal cancer (TCGA-ESCA) database: "primary therapy outcome" dataset (94 patients) and "OS event" dataset (162 patients). Figure 1 showed the overall approach of the study and Tables S1–S3 showed demographics of the above three datasets. The pCR group (11 patients) in the GSE45670 database served as the test group and the < pCR group (17 patients) was used as the control group. The CR group (74 patients) in the TCGA-ESCA "primary therapy outcome" dataset served as the test group and the < CR group (PD (10 patients) + SD (seven patients) + PR (three patients)) functioned as the control group. The Alive group (97 patients) served as the test group and the Dead group (65 patients) as the control group in the TCGA-ESCA "OS event" dataset. We performed a differential analysis in each of the three datasets, and statistical significance was determined by $|\log2 \text{ (fold change) }| > 1.5$ & $P < 0.05$ ($P < 0.05$ in GSE45670, adj. $P < 0.05$ in TCGA). Next, we obtained DEGs in three groups (Table S4), and screened the co-DEGs using a Venn diagram (Fig. 2A and Table S5). There were two co-DEGs identified (CTAG2 and MMP13). CTAG2 was highly expressed in the pCR group (log FC = 2.24) and CR group (logFC = 6.87), but low in the Alive group (log FC = −1.79) (Table S4). This is inconsistent with previous studies, which have found that patients sensitive to radiotherapy and chemotherapy have better prognosis and longer overall survival (OS) (*Zhao et al., 2020*; *Donington et al., 2003*; *Di Fiore et al., 2007*). MMP13 was highly expressed in the three experimental groups. Therefore, MMP13 might have some prognostic significance. Moreover, as shown in Figs. 2B & 2C, MMP1/9/10/13 were all highly expressed (log $|$ FC $| > 1.5$ & p.val $< 0.05$) in the pCR group, and MMP1/13 were regulated by the same transcription factors, AP-1 and NF-kB (*Vincenti & Brinckerhoff, 2002*). Both *Wen et al. (2014)* and *Zhang et al. (2020)* found that MMPs were significantly upregulated in pCR patients when they performed differential analysis on the GSE45670 microarray. Moreover, MMP genes were included in the prediction models they established, which indicated that MMPs are important factors affecting CRT sensitivity of ESCC. Thus, we conducted a follow-up study on the MMPs.

### Signaling pathways involved in DEGs and analysis of immune cell infiltration

To investigate the signaling pathways or immune regulation that MMPs might be involved in, we explored the KEGG database. MMP1/3/9/13 was found to participate in the IL-17 signaling pathway. Since IL-17 is related to cancer initiation, progression, and immunotherapy (*Wu et al., 2015*; *Chen et al., 2011*), this pathway was selected for further analysis. In Fig. S1, MMP13 and MMP1/3/9 are located downstream of the production of IL-17A/F by Th17 cells. MMPs are directly induced by IL-17 *via* activation of AP-1, NF-kB,

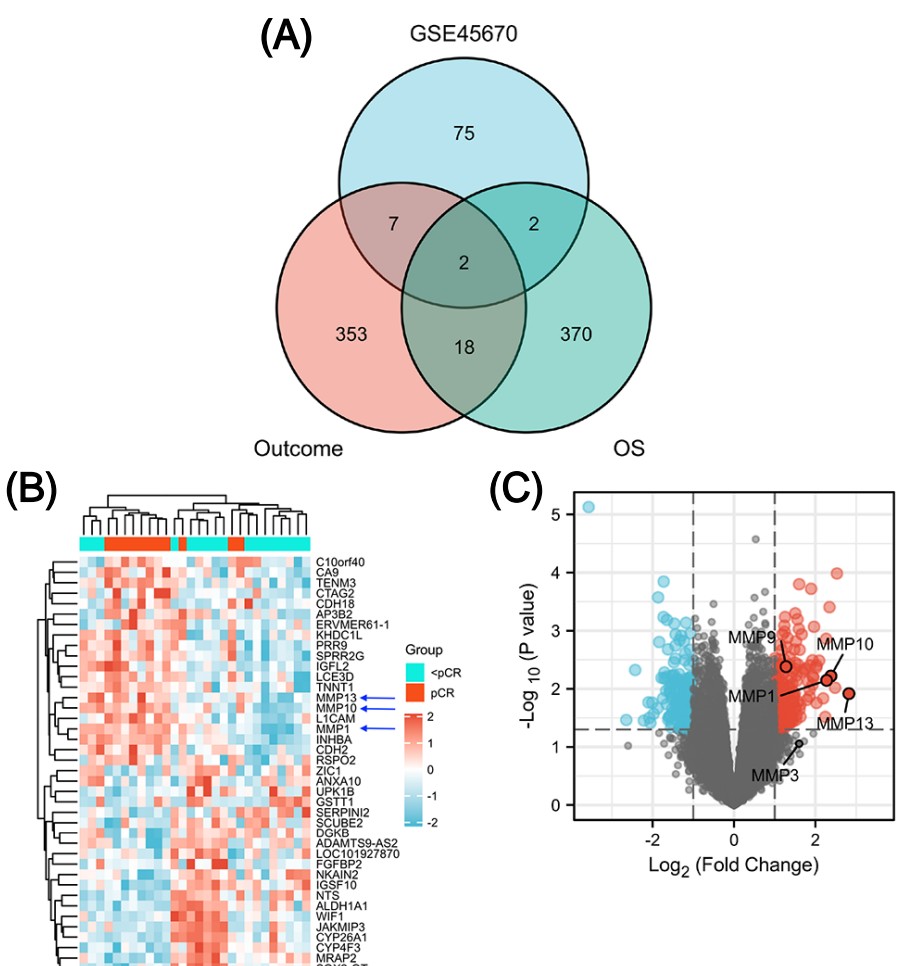

**Figure 2** **Identification of differentially expressed genes (DEGs).** Venn diagram shows two common DEGs in three datasets (A). The volcano plot and heat map show that MMPs are highly expressed in the pCR group (B, C). The expression scale in the heatmap represents the values of gene expression after the zscore conversion, that is, each value is subtracted from the row mean and then divided by the standard deviation. MMPs, matrix metalloproteinases.

and C/EBP-β *via* MAPK and ERK. Next, using ssGSEA algorithm, we performed infiltration analysis of 24 immune cells in TCGA-ESCA database (*Hänzelmann, Castelo & Guinney, 2013*; *Bindea et al., 2013*). Figure 3A shows that the expression of MMP13 is negatively correlated with the immune score of Th17 cells ($R = -0.600$, $P < 0.001$). Figure 3B shows that in the two groups grouped by the median of gene expression, the enrichment score of Th17 cells was lower in the group with high MMP13 expression. And the correlation coefficient between MMP13 and Th17 was the highest. Figure 3D shows that the expression of MMP9 is also negatively correlated with the immune score of Th17 cells ($R = -0.197$, $P < 0.05$). Although there was also a negative correlation between MMP1 and Th17 cells (Fig. 3C, $R = -0.065$), this correlation was not significant ($P > 0.05$). Therefore, a negative correlation could be observed between the downstream MMP genes and upstream IL-17A

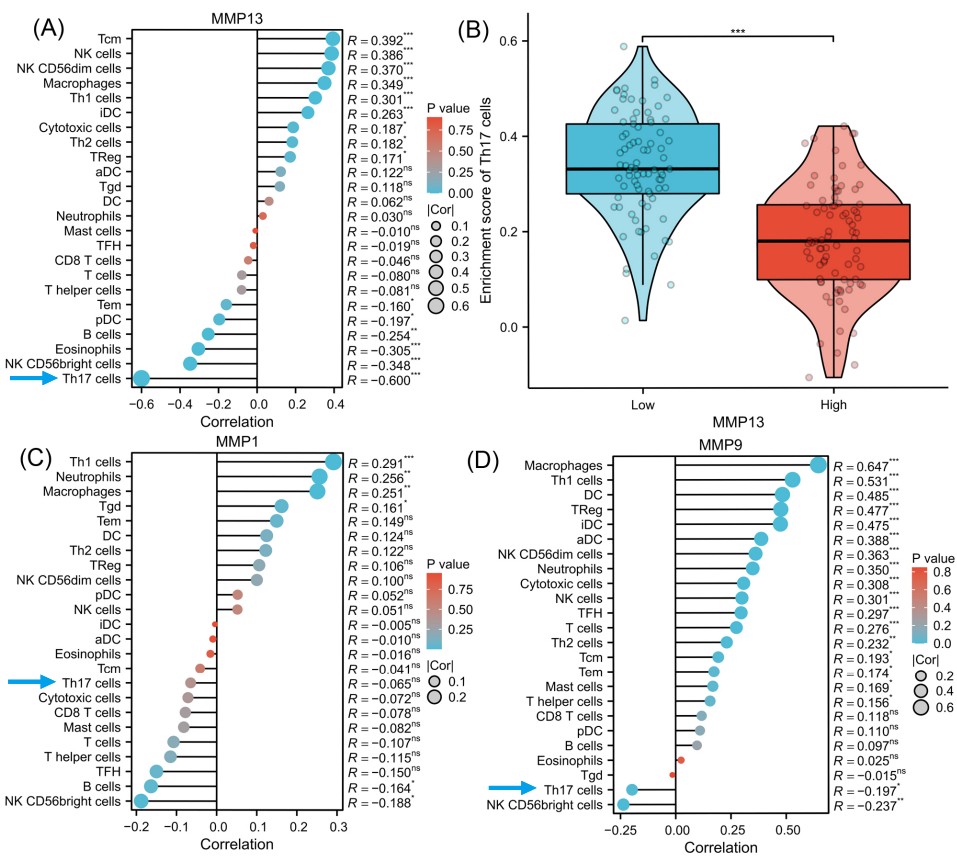

**Figure 3** **Immune infiltration analysis of MMPs.** The lollipop plot shows the correlation between MMP13 and immune cells (A). The violin plot shows the comparison of TH17 cell enrichment scores in MMP13 high and low expression groups (B). The lollipop plot shows the correlation between MMP1 and immune cells (C). The lollipop plot shows the correlation between MMP9 and immune cells (D). Lollipop plots are used to visualize the correlation of 1 gene and multiple cell fractions. Both the size of the circle and the height of the bar represent the degree of correlation, and the depth of the color represents the size of the $p$-value. *, $P < 0.001$. **, $P < 0.01$. ***, $P < 0.001$.

gene in the IL-17 signaling pathway, which is an interesting phenomenon. To provide additional confidence in the accuracy of immune cell type assignments, we used another deconvolution algorithm, MCP-counter, to calculate the correlation between MMPs and immune scores in the TCGA-ESCA dataset. Figure S3 shows that MMP13 is negatively correlated with B cells, and MMP13/9 are positively correlated with macrophages, dendritic cells, NK cells, neutrophils, and cytotoxic cells ($p < 0.01$). This is consistent with the results calculated by the ssGSEA algorithm, providing further validation for our immune cell type deconvolution.

## Gene Set Enrichment Analysis (GSEA)

To understand why the downstream genes MMPs was negatively correlated with the upstream gene IL-17A in esophageal cancer, we next performed GSEA analysis on all DEGs of CR/pCR *versus* <CR/ <pCR to explore signaling pathways that might explain

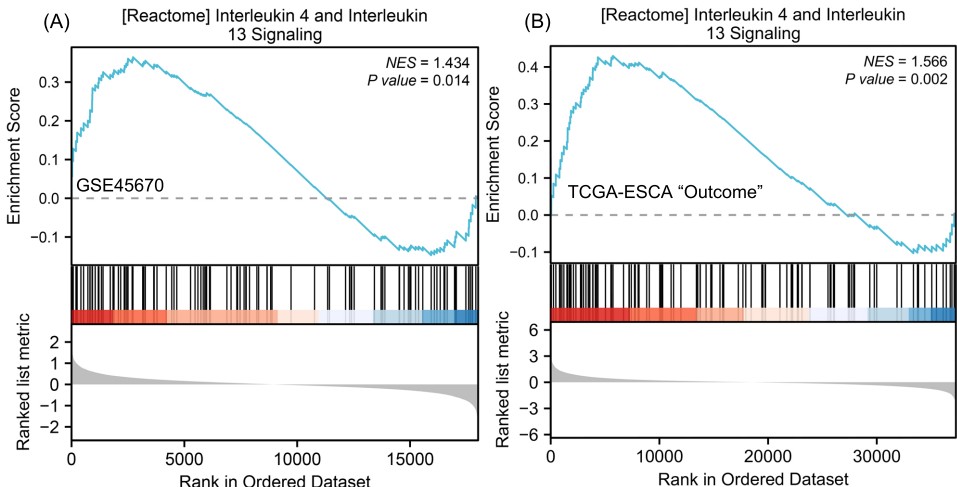

**Figure 4**  GSEA classic visualization of IL4/IL13 signaling pathway from two datasets. (A) The enrichment score of this pathway in the GSE45670 dataset (pCR group is the experimental group, <pCR group is the reference group). (B) The enrichment score of this pathway in TCGA-ESCA "primary therapy outcome" (CR group is the experimental group, <CR group is the reference group). In the upper part of each plot, if the NES is positive, the peak appears on the left (head enrichment) (high expression group enrichment), and the core molecules in the gene set are mainly concentrated in the left high expression group. Each vertical bar in the middle part represents a molecule in the gene set. The lower part represents the visualization of the normalized value of each molecule. CR, complete remission; <CR, CR not achieved; NES, Normalized Enrichment Score; pCR, pathological complete response; <pCR, pCR not achieved.

the correlation. The CR/pCR group was the experimental group, and the <CR/ <pCR group was the reference group. The criterion for significant difference is |NES | >1, NOM $p < 0.05$, FDR $q < 0.25$ in the enrichment of MSigDB Collection (c2.cp.biocarta and hall. v6.1 symbols). We used all of DEGs from TCGA-ESCA "primary therapy outcome" dataset (56,494 genes) and GSE45670 dataset (21,655 genes) to perform GSEA analysis respectively, and there are 17,639 co-DEGs in the two datasets (Table S4). We first obtained two enrichment analysis results (Table S11), and then screened for common signaling pathways in which the same MMPs were involved. We found that the IL-4 and IL-13 signaling pathways in the Reactome database were highly expressed in both the pCR and CR groups (Fig. 4, Table S6). According to KEGG IL-17 signaling pathway (Fig. S1), IL-4/5/13 and CCL11 can inhibit Th17 cells and promote a Th2 cell response, thereby reducing the level of IL-17A expression. In the inhibition of the Th17 differentiation pathway, IL-4 is secreted by Th2 cells and can reduce RORC (ROR $\gamma$t), which acts upstream of IL-17A and promotes IL-17A gene expression (Fig. S2). Therefore, MMPs are highly expressed in CRT-sensitive ESCA patients, and MMPs are negatively correlated with IL-17A, which may be related to the high expression of IL-4/IL-13 signaling pathway.

## Association of between MMP13, IL-4, RORC, and IL-17A with the clinicopathological parameters of ESCC patients

To verify potential correlations of the signaling proteins that exist in TME, we performed immunohistochemistry (IHC) to evaluate the four representative genes expression in

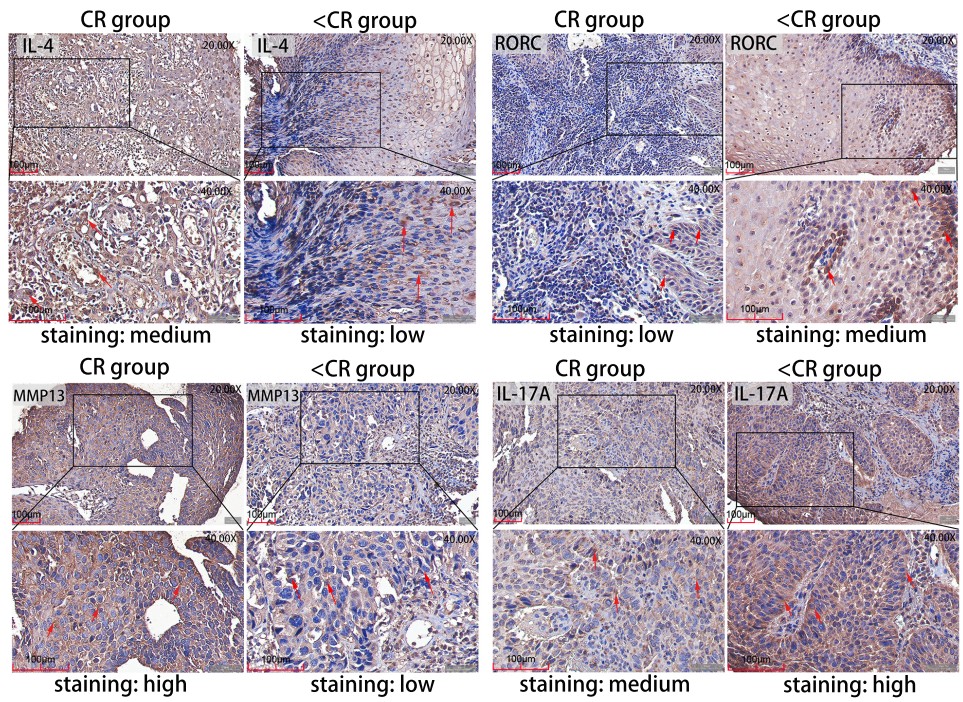

**Figure 5 Representative IHC images.** IHC staining evaluated the level of MMP13 and IL-4 and RORC and IL-17A protein expression in CR and <CR cancer tissues. Compared with <CR group, MMP13 and IL-4 protein staining were darker in CR group, but RORC and IL-17A protein staining were lighter. Original magnification: 20× and 40× (inset panels). IHC, immunohistochemistry; CR, complete remission; <CR, CR not achieved.

pretreatment ESCC biopsies, which were from six patients who received CRT and achieved complete remission (CR: three patients) or not (<CR: 3 patients). The IHC staining of the four proteins in CR and < CR cancer tissues are shown in Fig. 5. Combined with the IHC mean optical density values IOD/AREA of pathological sections in each of the two groups compared with independent $t$-test (Fig. 6), we found that the CR group expressed a higher level of MMP13 and IL-4 compared to the <CR group. However, the CR group displayed significantly lower levels of RORC and its activated downstream gene IL-17A expression than its counterparts. We also selected MMP3 (another MMP gene involved in IL-17 signaling pathway) and RORA (another upstream gene promoting IL17A expression) to perform IHC and a mean optical density analysis (Figs. S4 and S5). The CR group showed higher MMP3 levels (marginally significant) and lower RORA levels ($P < 0.01$) than the <CR group.

## Association between IL-17A expression, clinical variables and prognosis analysis

After verifying the negative correlation between the key differential gene MMP13 and its upstream gene IL-17A, we explored the clinical relevance of IL-17A. An analysis of 173 ESCA samples with IL-17A expression data together with all the patient characteristics from TCGA database was conducted to investigate the role and significance of IL-17A. A

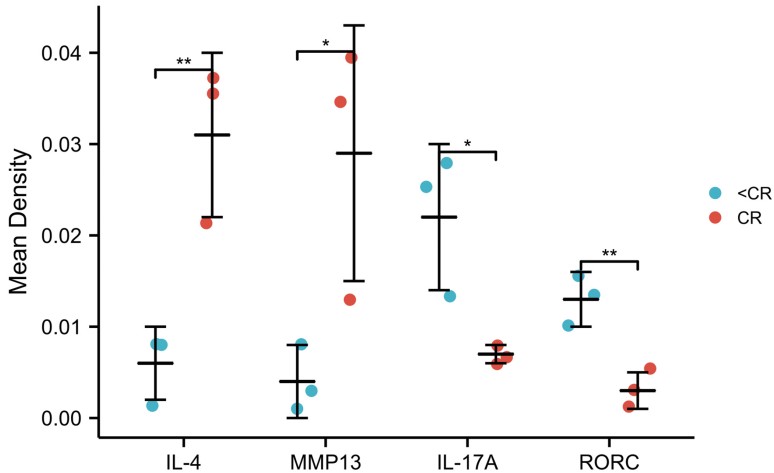

**Figure 6** IHC mean optical density values IOD/area of the CR and <CR groups. Compared with the <CR group, the CR group had higher mean optical densities of MMP13 and IL-4, but lower mean optical densities of RORC and IL-17A. *, $P < 0.001$. **, $P < 0.01$. IHC, immunohistochemistry; IOD, integrated optical density; CR, complete remission; <CR, CR not achieved.

Wilcoxon rank-sum test was used to compare gene expression values. Figures 7A & 7B show that IL-17A overexpression was associated with poor primary therapy outcomes (PD & SD & PR *vs.* CR, $P = 0.034$) and OS event (Dead *vs.* Alive, $P = 0.008$), which was in contrast to MMP13 expression. In addition, based on Kaplan–Meier analysis and log-rank test, a univariate survival analysis of IL17A was conducted in TCGA-ESCA database (Fig. 7C). The patients were separated into low and high subsets based on a gene expression value of 0-20 compared to 80-100. A higher level of IL-17A was associated with lower OS ($P = 0.046$, HR = 2.15). Based on these results, a high level of IL-17A in ESCAs was associated with a lower CRT sensitivity and a higher mortality rate.

## Immune-related prognostic signature construction

We found that high expression of IL-4/IL-13 signaling pathway may be related to the negative correlation between MMP13 and IL-17A, and both MMP13 and IL-17A have clinical significance. Therefore, establishing a prognostic signature of ESCA based on these signaling molecules and immune genes is necessary to guide early patient management. We followed up the 28 patients in the GSE45670 until the cut-off in December 2021 (Table S7). A median follow-up time of 3,746 days was observed (95% CI [3,649–4,651] days). The total number of deaths was 13, and the proportion of censored data was 53.6%. GSE45670 differential analysis showed that the differential expression folds of MMP1/3/9/10/13 were large (log FC > 1). According to Fig. S1, we found that MMP1/3/9/13 is downstream of IL-17A in the Th17 cell differentiation pathway, and IL-4/IL-13/CCL11 can inhibit the response of Th17 cells. Moreover, the core enriched molecules of IL-4/IL-13 signaling pathway include MMP1/2/3/9/13 and IL-4/IL-13 (Table S6). So these 10 immune-related signaling molecules (MMP1/2/3/9/10/13, IL-4/IL-13, CCL11, IL-17A) are the core genes of our analysis, and they may be related to the negative correlation between MMPs and IL-17A.

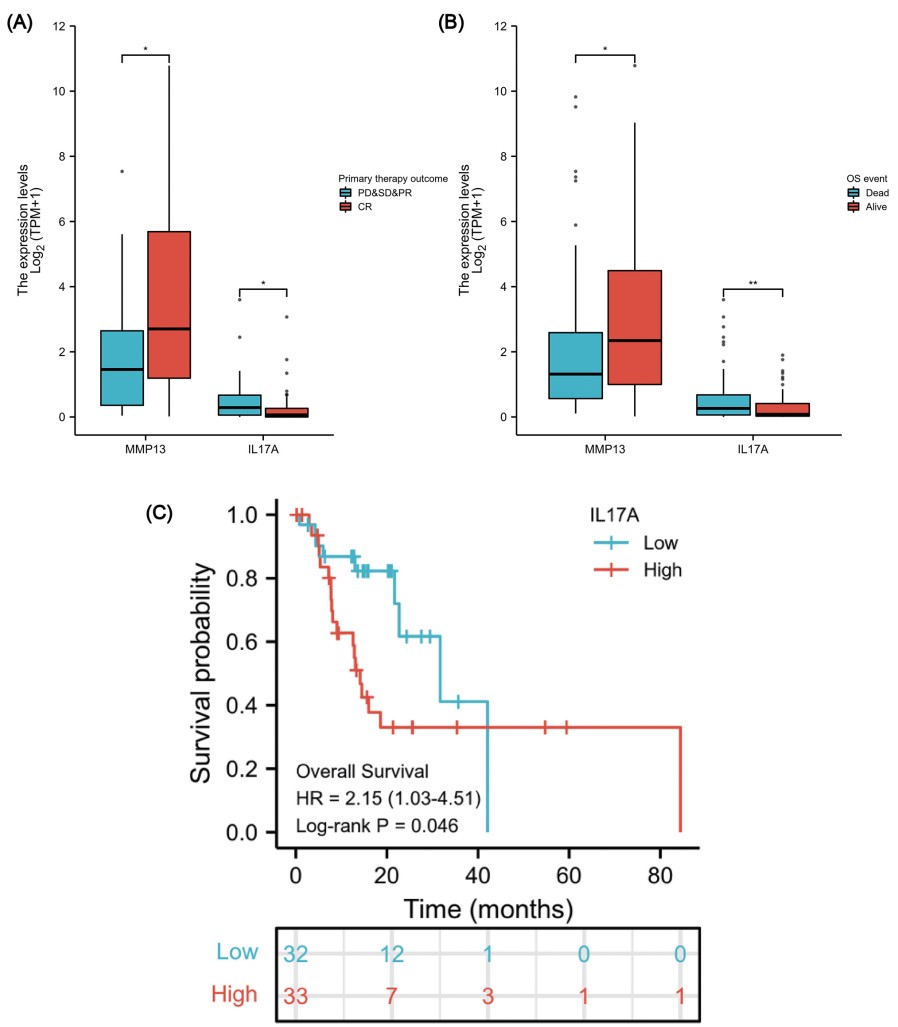

**Figure 7** **Association between IL-17A Expression, Clinical Variables, and Prognosis.** The difference in MMP13 and IL17A expression in the "primary therapy outcome" dataset (A), the difference in MMP13 and IL17A expression in the "OS event" dataset (B), and survival analysis of IL-17A in the TCGA-ESCA database (C). *, $P < 0.05$. **, $P < 0.01$. TCGA, The Cancer Genome Atlas; ESCA, esophageal cancer.

For stratifying patients' clinical outcomes rapidly and efficiently, the 10 genes were analyzed using the Lasso regression algorithm to establish a prognostic signature in the GSE45670 cohort (Fig. 8A). Using cv.glmnet function, a tenfold cross-validation was performed to determine the lambda minimum that gives the minimum cross-validated error. In a cohort of 28 cases, lambda Min was used as the cutoff, and there were five variables (MMP3, MMP13, IL-4, IL-13, and IL-17A) with non-zero coefficients. A classifier was established with the formula:

$$\text{risk score} = \sum_i \text{Coefficient}\left(\text{hub gene}_i\right) * \text{mRNA Expression}\left(\text{hub gene}_i\right)$$

$$= (-3.17091E - 05) \times MMP3 + (-3.95119E - 05) \times MMP13$$

$$+ 0.050168703 \times \text{IL4} + 0.015660841 \times \text{IL13} + 0.00261372 \times \text{IL17A}.$$

To evaluate the predictability of the formula with survival data, the time-dependent receiver operating characteristic (ROC) curve for risk scores was used (Figs. 8B and 8C), which suggested that the area under the curve (AUC) for 1/2/3/4/5 years was all greater than 0.70, with the 5-year AUC being the largest. A cutoff value of 0.987 was used to correctly classify nine of 10 as an OS less than 5 years with 90% sensitivity. Additionally, a specificity of 72.2% was achieved for 13 out of 18 correct classifications of $\geq$ 5 years. We achieved an overall accuracy of 78.6% (22 of 28) with an AUC of 0.850 for our signature (95% CI [0.709–0.991]). Using the median risk scores, twenty-eight cases were divided into two risk subsets: low and high, and we analyzed survival outcome as well as gene expression in the model according to the grouping (Fig. 8D). The findings showed that there were fewer death events (blue points) and more surviving patients (red points) in the low-risk group. Moreover, MMP3/13 and IL-4 were highly expressed, whereas there was low IL-17A expression. Using a Kaplan–Meier survival analysis, we were able to observe that low-risk individuals are significantly more likely to survive compared to high-risk individuals (Fig. 8E). The log rank test results indicated that the survival times of both groups significantly differed ($P = 0.003$, HR = 22.43), indicating a worse prognosis for the high-risk group.

### Validation of immune-related prognostic signature in TCGA cohorts

In order to verify whether this prognostic model has good prediction performance in external cohorts, we explored it in TCGA-ESCA database. There were 23 ESCC patients receiving both radiotherapy and chemotherapy, named the CRT cohort, and their median follow-up was 383 days (patients' information in Table S8). Due to the short follow-up, we could only obtain a 1-year ROC curve. As of 1-year, there were two deaths, two censors and 19 patients were still alive. Using the model, 15 of 21 samples were successfully identified with an accuracy of 71.4% and an AUC of 0.739 (Fig. 9A, 95% CI [0.538–0.941]). Using the optimal cut-off value 0.00325 of risk score, low-risk patients survived longer based on Kaplan–Meier analyses (Fig. 9B, $P = 0.041$, HR 6.59). However, because the number of events in the CRT cohort is too small, verification may be not strongly convincing. Therefore, we expanded the sample size in TCGA-ESCA database and validated the model in ESCC patients who had received radiotherapy or chemotherapy. There were 41 patients, named RorC cohort (patients' information in Table S9). The median follow-up was 391 days, so we can only plot a 1-year and a 2-year ROC curve. As of 1-year, there were two deaths, four censors and 35 patients were still alive; 27 of 37 samples were successfully identified with an accuracy of 73.0% and an AUC of 0.773 (Fig. 9C, 95% CI [0.621–0.924]). As of 2-year, there were five deaths, 31 censors and five patients were still alive; seven of 10 samples were successfully identified with an accuracy of 70.0% and an AUC of 0.695 (95% CI [0.250–1.141]). Using the optimal cut-off value 0.00325 of risk score, high-risk patients had shorter survival times (Fig. 9D, $P = 0.034$, HR 4.03, 95% CI [0.88–18.51]).

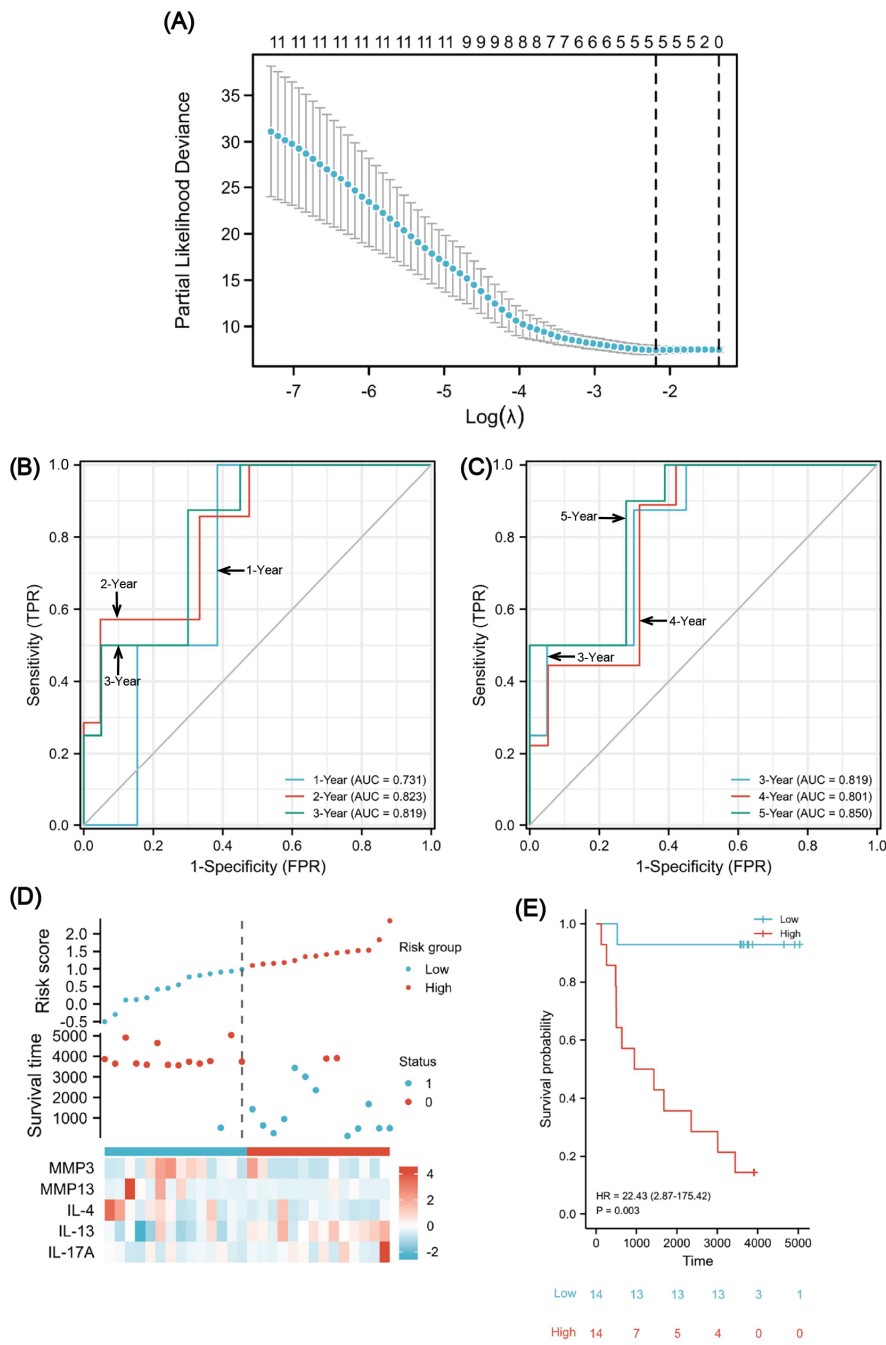

**Figure 8  A five-gene immune-related prognostic signature construction.** Lasso regression screened five variables to establish a prognostic model (A). The AUCs for 1/2/3/4/5 years was all greater than 0.70 (B, C). The risk factor diagram revealed that low-risk patients had a longer survival time (D). The Kaplan-Meier curves revealed a significantly higher survival probability for the low-risk group (E). AUC, area under curve.

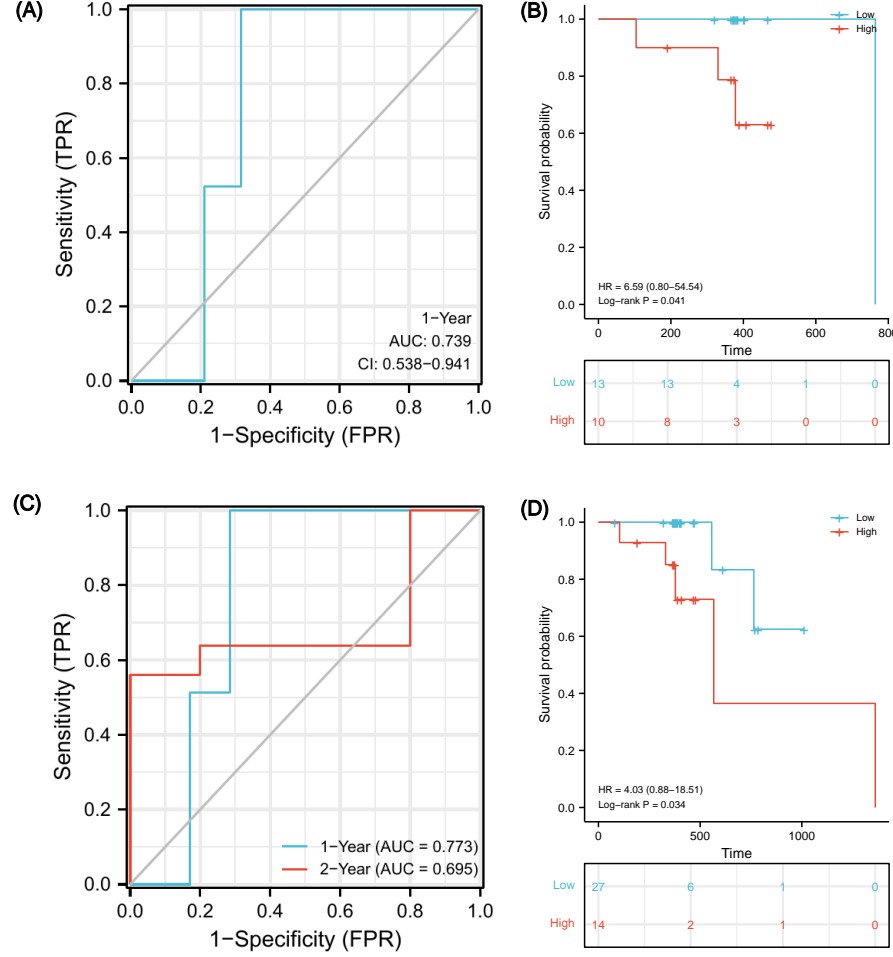

**Figure 9** **Validation of the prognostic signature in two external cohorts.** Time-dependent ROC curve at 1-year from the CRT cohort (A). Kaplan–Meier curves of high and low risk groups in the CRT cohort (B). Time-dependent ROC curves at 1- and 2-year from the RorC cohort (C). Kaplan–Meier curves of high and low risk groups in the RorC cohort (D). AUC, area under curve; CRT, chemoradiotherapy; ROC, receiver operating characteristic; RorC, radiotherapy or chemotherapy.

## DISCUSSION

The prognosis of ESCC in many countries remains extremely poor and it is difficult to predict which patients have a relatively optimistic outcome (*Huang et al., 2020*). Previous studies have found that patients who are sensitive to radiotherapy and chemotherapy have better prognosis (*Zhao et al., 2020*; *Donington et al., 2003*; *Di Fiore et al., 2007*). Therefore, it is important to study the molecular mechanisms that affect CRT sensitivity, and establish a prognostic model that can be used for therapeutic schedule selection. Recently, Th17 cells have become a hotspot because of their duality in differentiation and function. Researchers have found that Th17 lymphocytes in tumor tissue can promote the metastasis of lung cancer (*Salazar et al., 2020*). In addition, Th17 cells may serve as a new option for tumor adoptive cell therapy of melanoma due to their antitumor activity (*Bowers et al., 2017*).

Similarly, in the field of esophageal cancer, IL-17A can not only promote the invasion of tumor cells (*Liu et al., 2017*), but also induce anti-tumor immunity by recruiting and activating beneficial B cells and neutrophils (*Chen et al., 2017*; *Lu et al., 2016*). Therefore, it is necessary to further study immunomodulation and prognostic role of IL-17A.

MMPs are produced by cancer cells, CAFs, ECs and various immune cells in the TME (*Wang et al., 2022*). Two previous studies on GSE45670 microarrays found that MMPs were significantly upregulated in pCR patients, which indicated that MMPs are potential biomarkers of CRT sensitivity (*Wen et al., 2014*; *Zhang et al., 2020*). However, the researchers did not further explore the immune response regulation that MMPs may participate in. We found that MMP13 and other MMPs were highly expressed in the complete remission patients who received CRT and were associated with a good prognosis. MMP1/3/9/13 mapped on to the IL-17 signaling pathway (Fig. S1). It takes p38 MAPK and ERK1/2 activation for IL-17 to activate NF-kB, AP-1, and C/EBP-$\beta$ and to induce MMP-1 expression. Due to the similarity of TF binding sites in the promoters of these MMPs, MMP-2, -3, -9, and -13 are also induced by IL-17, which play an important role in remodeling tissues (*Cortez et al., 2007*). Interestingly, however, we found that MMPs were negatively correlated with the upstream gene IL-17A and Th17 cell infiltration. Moreover, IHC analysis verified that there was a higher expression of MMPs and lower IL-17A in ESCC patients who achieved CR after CRT, compared with the non-CR patients. Clinical studies have also suggested that a high Th17 cell frequency may be a strong predictor for poor prognosis in cancer (*Han et al., 2014*). Indeed, high levels of IL-17A have been shown to be independent predictors of poor survival in gastric cancer patients (*Zhuang et al., 2012*). However, multiple MMPs have protective effects in multiple tumor types. For example, in squamous cell carcinoma, the initial tumor growth rates of wild-type mice expressing MMP3 were lower than that of knockout mice (*McCawley et al., 2004*). MMP13 can also digest the dense extracellular collagenous matrix surrounding tumors, so as to increase the sensitivity of tumors to chemotherapy and enhance anti-tumor activity (*Long et al., 2016*). With the aim to verify whether MMP family genes affect CRT sensitivity, the genetic alterations in long-term fractionated radiation-acquired non-radioresistant *vs.* resistant ESCA cells were investigated in GSE61620 dataset from GEO. We set radiation sensitive ESCC cells (GSM1509454, GSM1509455, GSM1509456) as the experimental group, and the non-sensitive cells (GSM1509457, GSM1509458, GSM1509459) as the control group for DEGs analysis (Table S10). We found that MMP3 was highly expressed in radiosensitive cells (log FC = 1.011, $p$.val = 0.017). Like MMP13, MMP3 is another downstream MMP family gene located in the IL-17 signaling pathway (Fig. S1). MMP3 and MMP13 are both genes in our prognosis signature based on the IL-17 signaling pathway.

Moreover, the GSEA enrichment analysis suggested that the IL-4 and IL-13 signaling pathways in the Reactome database were also highly expressed in pCR/CR patients with high MMP expression (Fig. 4). Two chains of IL-4R and IL-13R are shared (IL-4Ra and IL-13Ra1), which signal through the JAK/STAT pathway (*Suzuki et al., 2015*). Based on the Th17 cell differentiation pathway, high IL-4 expression may up-regulate GATA3 through the JAK/STAT signaling pathway, blocking DNA expression of RORC, thereby reducing IL-17A secretion (Fig. S2). Therefore, the negative correlation between MMPs and IL-17A

in ESCA patients may be related to the high expression of IL-4/IL-13 signaling pathway. We verified higher levels of IL-4 and MMP expression and lower levels of IL-17A and RORC expression using IHC in the pretreatment biopsies of ESCC patients from the Ordos region who achieved CR after CRT compared to those who were non-CR. Both the reports by *Cochaud et al. (2013)* and *Merrouche et al. (2016)* found that ERK activation and HER1 phosphorylation mediated by IL-17 could promote the drug resistance of taxane-based chemotherapy and resistance of tyrosine kinase inhibitors. Researchers have found that low-dose radiotherapy of the tumor bed may increase the expression of IL-17A and promote invasive growth of the tumor by producing IL-6 and TGF-$\beta$ (*Lee et al., 2014*). Based on these results, IL-17A may promote the resistance of tumor cells to CRT, leading to tumor progression, recurrence, and metastasis. In conclusion, MMPs are highly expressed in CRT-sensitive ESCA patients, and MMPs are negatively correlated with IL-17A, which may reduce the efficacy of CRT. This correlation may be related to the high expression of IL-4/IL-13 signaling pathway. Therefore, it may be meaningful to construct clinical prognostic signatures based on these signaling molecules and immune genes.

In the TCGA-ESCA database, IL-17A prognostic significance was validated. The OS of low IL-17A group was better than the high group, which indicated that the level of IL-17A expression is negatively correlated with patient prognosis. Moreover, our findings suggested that after receiving primary therapy, including chemotherapy or radiotherapy, patients achieving CR had lower IL-17A and higher MMP13 levels compared with non-CR patients. Previous studies have also revealed that elevated Th17 cells secreted IL-17A to promote cell growth and resistance to antitumor therapy in tumors (*Bi et al., 2016*; *Zhong et al., 2019*).

Finally, we constructed a five-gene immune-related signature based on the MMP family and IL-17 signaling pathway. In GSE45670 training cohort, patients with ESCC could be predicted to survive for a long time with this clinical efficacy evaluation model, and the predictive accuracy for the five-year OS was the highest. Additionally, the risk score formula could distinguish high-risk from low-risk patients. There was a significant survival advantage in the low-risk score subset based on K-M analyses. To verify that our signature is universal in the ESCC population who received CRT, external validation was performed on two TCGA-ESCA cohorts. In the CRT cohort, 1-year AUC = 0.739 (0.538–0.941), which indicates that the model has certain accuracy in predicting 1-year OS. K-M survival analysis also shows that this formula can well distinguish high- or low-risk populations with obviously different survival (HR = 6.59, $P = 0.041$). In order to expand the sample size and increase the credibility of model prediction, we explored in the RorC cohort who received chemotherapy or radiotherapy. Similarly, 1-year AUC = 0.773 and 2-year AUC = 0.695, which indicate that the model has robust accuracy in predicting 1-year OS but low accuracy in predicting 2-year OS. Considering some patients in this cohort did not receive both radiotherapy and chemotherapy, this population heterogeneity may lead to the reduced prediction efficiency of this model. Notably, it can also well distinguish high- or low-risk subsets. The survival for high-risk patients seemed to be shorter (HR = 4.03, $P = 0.034$). Therefore, our immune signature's predictive power was creditable and stable across the training and external validation cohorts, with AUCs exceeding 0.7.

The study was associated with some limitations. First, although we found a negative correlation between MMPs and IL-17A, this study is observational in nature, and the molecular mechanism of IL-4/13 regulation are speculative based on existing literature. Further experimental studies are needed to fully elucidate the underlying biological mechanisms driving this negative correlation. In the future we will perform *in vivo* and *in vitro* experiments to verify the exact role of these markers in CRT sensitivity in ESCC patients. Secondly, due to the difficulty of collecting large biopsy or resection specimens, we were only able to perform IHC validation on a small sample size, which was insufficient to quantify immune cells for a more robust assessment. Thirdly, no large sample size validation cohort with sufficient OS data exists to increase the reliability of the model. The collection of clinical samples requires a few years. Therefore, we will accumulate more samples with CRT sensitivity to extend these analyses a larger cohort for further verification.

## CONCLUSIONS

An immune-related signature predicting long-term survival after CRT for ESCC patients was constructed based on the MMP family and IL-17 signaling pathway. In the knowledge of authors, the study firstly suggested immune cell infiltration and signaling pathways involved in the prognostic model genes and reported a possible mechanism of their mutual regulation. MMPs are highly expressed in CRT-sensitive ESCA patients, and MMPs are negatively correlated with IL-17A, which may reduce the efficacy of CRT. This correlation may be related to the high expression of IL-4/IL-13 signaling pathway. Moreover, our predictive model effectively stratified ESCC patients into high-risk and low-risk groups with different OS outcomes. This may have practical implications for clinicians, enabling more appropriate early prognostic management of high-risk patients. It is important to note that the clinical significance of our findings should be interpreted with caution, as further validation and prospective studies are required. In conclusion, our study contributes to the understanding of immune-related factors in ESCC and their association with long-term survival after CRT. The immune signature developed in this study holds promise as a prognostic tool but requires further validation. These findings provide insights into potential therapeutic targets and prognostic biomarkers, and more research is needed to elucidate their clinical significance.

## ACKNOWLEDGEMENTS

For editing our manuscript, we gratefully acknowledge Elixigen Company (Huntington Beach, California).

### Funding

This work was supported by the 2022 Science and Technology Plan Projects of Ordos Municipal Bureau of Science and Technology (No. 2022YY014), the Natural Science Foundation of China (No. 81874220) and the Natural Science Foundation of Guangdong Province (No. 2022A1515012483 and No. 2020A1515010030). The funders had no role in study design, data collection and analysis, decision to publish, or preparation of the manuscript.

### Grant Disclosures

The following grant information was disclosed by the authors:
2022 Science and Technology Plan Projects of Ordos Municipal Bureau of Science and Technology: 2022YY014.
Natural Science Foundation of China: 81874220.
Natural Science Foundation of Guangdong Province: 2022A1515012483, 2020A1515010030.

### Competing Interests

The authors declare there are no competing interests.

### Author Contributions

- Zewei Zhang performed the experiments, analyzed the data, prepared figures and/or tables, authored or reviewed drafts of the article, and approved the final draft.
- Shiliang Liu performed the experiments, analyzed the data, authored or reviewed drafts of the article, and approved the final draft.
- Tiantian Gao analyzed the data, authored or reviewed drafts of the article, and approved the final draft.
- Yuxian Yang analyzed the data, authored or reviewed drafts of the article, and approved the final draft.
- Quanfu Li conceived and designed the experiments, authored or reviewed drafts of the article, and approved the final draft.
- Lei Zhao conceived and designed the experiments, authored or reviewed drafts of the article, and approved the final draft.

### Human Ethics

The following information was supplied relating to ethical approvals (*i.e.*, approving body and any reference numbers):

The study protocol and the use of human samples had the ethics committee approval of Sun Yat-sen University Cancer Center and Ordos Central Hospital (ethics numbers: SL-B2022-054-01).

### Data Availability

Data are available at Zenodo:

Zewei Zhang. (2023). A novel immune-related prognostic signature based on Chemoradiotherapy sensitivity predicts long-term survival in patients with esophageal squamous cell carcinoma. [Data set]. Zenodo. https://doi.org/10.5281/zenodo.8146998.

## Supplemental Information

Supplemental information for this article can be found online at http://dx.doi.org/10.7717/peerj.15839#supplemental-information.

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
