# Peer review of "A novel immune-related prognostic signature based on Chemoradiotherapy sensitivity predicts long-term survival in patients with esophageal squamous cell carcinoma"

_PeerJ, doi:10.7717/peerj.15839_

## Round 0.1 · original submission · Major Revisions

Please follow carefully reviewers suggestions. This manuscript needs to be improved properly.

·

Basic reporting

The manuscript is professionally written. It is clear and easy to follow. Professional English is utilized throughout.
Although the introduction has some background information, it lacks to provide sufficient information to build the premise of the manuscript. The introduction can be written more extensively to define the current situation, elucidate the gaps in understanding in terms of the disease and the prognosis and pose a problem which is addressed in the experiments performed in the manuscript. Even though these points are touched upon, it feels truncated and leaves room for improvement. Also, some key publications have been left out from the introduction which will provide enough context to build the premise of the manuscript. ( Chen, C. L., Wang, Y., Huang, C. Y., Zhou, Z. Q., Zhao, J. J., Zhang, X. F., ... & Xia, J. C. (2018). IL-17 induces antitumor immunity by promoting beneficial neutrophil recruitment and activation in esophageal squamous cell carcinoma. Oncoimmunology, 7(1), e1373234.; Jiao, X. L., Chen, D., Wang, J. G., & Zhang, K. J. (2014). Clinical significance of serum matrix metalloproteinase-13 levels in patients with esophageal squamous cell carcinoma (ESCC). Eur Rev Med Pharmacol Sci, 18(4), 509-515.; Lu, L., Weng, C., Mao, H., Fang, X., Liu, X., Wu, Y., ... & Liu, G. (2016). IL-17A promotes migration and tumor killing capability of B cells in esophageal squamous cell carcinoma. Oncotarget, 7(16), 21853.; Liu, D., Zhang, R., Wu, J., Pu, Y., Yin, X., Cheng, Y., ... & Zhang, J. (2017). Interleukin-17A promotes esophageal adenocarcinoma cell invasiveness through ROS-dependent, NF-κB-mediated MMP-2/9 activation. Oncology Reports, 37(3), 1779-1785.)
Some areas of improvements also include including interleukins to the keywords (line 43), explaining some abbreviations used like IL-17 A/F (line 63), explaining fundamental differences between pCR and CR (line 33 & 72), clarifying nomenclatures like <CR, etc.
The article structure abides by the rules put forward by PeerJ. The article flows seamlessly and is easy to follow overall. Raw data and sufficient supplemental information is provided to understand the rationale and experiment designs.

Experimental design

The scope of the article and the aim of the research is very well detailed in the manuscript. The gaps in understanding is well defined and the relevance of the work is also elucidated very well.
Method description is done in detail for most part. Secondary antibody information, substrate information and washing and counterstaining step details are missing from the IHC protocol (line 103). In line 107, the expression measured using IHC is protein expression, hence the name of the proteins should not be italicized (italics is only used for gene names).
The experimentation was performed thoroughly and the data provided concurs with the description.

Validity of the findings

The investigation is very thoroughly documented and in depth analysis is done to ensure that the data is robust and statistically sound. The discussion is well articulated and the conclusion is well drawn out. Each of the data points are discussed and how it leads to building of the model was also examined rigorously.
Only caveat of the conclusion is the inclusion of IL-4 and IL-13 in the model. This was touched upon in the discussion, that the molecular mechanism of IL-4/13 regulation and MMP is not clear and was discussed as a potential shortcoming of the work.

Additional comments

Introduction needs to be expanded and bolstered with more relevant and recent references. Some proofreading of methods is necessary. Detailed method can be submitted as a supplementary document. Either more experiment needs to be designed with IL-4 and IL-13 to establish them as a part of the model or redesign of the model is required to not incorporate them in the current capacity.
English translation of the approval letter needs to be provided.

Reviewer 2 ·

Basic reporting

The article requires extensive language editing and proofreading. The presentation of data, text and figures is not very easy to follow. I commend the authors for referencing the databases that have been used but they need to provide more specifics and details in the methods section. Several figures require reformatting and are missing critical details from their legends. Fig 1- several typographical errors that need to be fixed. Fig 2- unclear what the expression scale is for. Fig 3 appears to be a screenshot of the KEGG IL17 pathway. The pathway cannot be pasted from the webpage as is but should be redrawn in powerpoint with relevant elements of the pathway highlighted. Fig 4- it is unclear why the p value scale goes from 0 to 0.75 making it difficult to decipher which subsets are in fact significant (<0.05). Fig 5- images are missing the size bar.

Experimental design

The research question is well defined but the title overstates the conclusions. The study uses correlation as a tool for understanding association but implies mechanistic causation as part of their conclusions for the same. The rationale for connecting MMP13 to IL17 specifically is very poorly defined. The methods applied are inadequately described making it difficult to judge the veracity of their claims. For instance, it is unclear which algorithms/ pipelines were used to infer immune cell frequencies. Moreover, the authors have performed actual IHC validation in a very small number of sample (n=3 each for CR and <CR). It is not unreasonable to extend these analyses the whole cohort and quantify immune cells using IHC.

Validity of the findings

Conclusions and the title vastly overstate the inferences that can be made from the analyses.

Reviewer 3 ·

Basic reporting

There are typing errors across the paper. The authors should carefully proofread the text and correct it. There is scope for improvement in the figures and figure legends could be made more informative. Legends are also missing on panels and should be added as necessary. Authors should ensure any interpretation in the text should be shown as a figure (e.g. quantification of pathway analysis).

Experimental design

1) There are several issues with the IHC data presented. One of the major problems is the lack of markers to identify cell types of cells on the slides. The protein expression quantified cannot be interpreted unless there is an additional marker for tumor cells, Th17 cells at the least since that is the focus of this study.

2) The initial choice of the 10 genes for the signature in Fig 8 seems arbitrary. What about other genes in these pathways?

3) Th17 cells seem to be the focus of the study based on literature. Can the authors elaborate on why this subset is more important than any other? MMPs are also expressed by several cell types in the stroma. The authors need to account for the multiple possible sources of MMPs in the samples.

4) Can the authors show the difference in pathway scores for MMP high vs MMP low samples? (Lines 313-315). There is a claim of high pathway expression, but the figures do not show any pathway score quantification.

5) Based on the volcano plots in Fig 2C, there are several genes that are more differentially expressed than the MMP genes. Could the authors explain why those were not a focus of this study?

6) The ranked gene set enrichment results should be provided as a table, not just the common gene sets selected as shown in Table S6.

Validity of the findings

1) The authors should use at least one other deconvolution method (like Cibersort) to confirm the immune cell type deconvolution with the cell type scores assigned to the samples. Do the two scores correlate with each other?

2) A negative correlation is reported by the authors between MMPs and IL-17A. Could the authors elaborate on the biological significance of this since this is unexpected?

3) Lines 326-330 have a lot of mechanistic implications without any validation experiments. Citing other papers is insufficient to establish a strong link between these markers and their roles in this context.

---

## Round 0.2 · accepted · Accept

All the Reviewers are satisfied of the revisions performered. Thank you.

·

Basic reporting

Errors corrected.

Experimental design

Addition of method and antibody information is satisfactory.

Validity of the findings

I am satisfied with the conclusion drawn from the data. Acknowledgement by the authors that further work is needed to test the hypothesis and validity of the model proposed is satisfactory.

Additional comments

The revised version of the manuscript addresses all the points brought up during the review. The rebuttal letter by the authors also addresses these issues point by point.
The publications are cited as mentioned in the review. Typographical errors and grammatical mistakes are also corrected. The method description is modified and additional description added to meet the standard of the journal. It is evident that extensive proofreading has been carried out.
As for the modification of the model, I am satisfied with the justification of the inclusion of IL-4 and IL-13 in the model proposed. The referencing of the GSEA data strengthened the rationale of this addition.
Overall I feel the manuscript is suitable for publication after these changes.

Reviewer 2 ·

Basic reporting

The authors have significantly improved their reporting of methods, legends and figures in general. No further changes required.

Experimental design

No comment

Validity of the findings

Satisfied with the additional explanation in the text and methods for the revised manuscript

Additional comments

No comment

Reviewer 3 ·

Basic reporting

Significant changes have been made to the manuscript to make it easier to read. The figures and tables are also improved in the revised version.

Experimental design

No comment

Validity of the findings

No comment

Additional comments

The authors have addressed my comments sufficiently and made the necessary changes to the text and figures.